# Training forecast to football athletes using Hopfield neural networks based on Markov matrix

**Hongxing Peng** [1]*, **Li Li** [2], **Long Cheng** [1]

1 Guangdong University of Technology, Guangdong, China, 2 Guangdong Polytechnical Normal University, Guangdong, China

* hongxingpenggd@163.com

## Abstract

This paper proposes a neural network based on the Markov probability transition matrix to predict the training performance of football athletes. Firstly, seven training indicators affecting the training performance are designed by the Event-group training theory. Then, a discrete Hopfield neural network is employed according to the seven training indicators. To improve the forecast ability of the discrete Hopfield neural network, the Markov probability transition matrix is used to calculate the activation probability of neurons. Finally, experimental results indicate that the proposed model defeats against the competitors in the forecast of training performance of football athletes. And the proposed model can find the major training indicators that have direct effects on the training performance, which can provide scientific suggestions for coaches to customize training plans. We demonstrate that the seven training indicators can sufficiently evaluate the effectiveness of training plans in the improvement in terms of training performance for football athletes.

## 1. Introduction

With the continuous development of science and technology, sports training has gradually become more scientific and systematic. Predicting athlete training results has become a focus of attention for coaches and athletes, since the training results are of great significance for developing reasonable training plans and adjusting training strategies.

Regarding the prediction of athlete training results, existing research has focused more on the following aspects: (i) the relationship between physical fitness training and specialized performance. (ii) The role of inspection indicators in predicting athlete performance. (iii) Models for training load and physical fitness status. However, existing research has not fully explored other factors influencing athlete training, such as psychological quality, technical and tactical level, etc.

To efficiently and accurately explore factors influencing athlete training, these prediction models of focusing on athlete performance are widely taken. In fact, these prediction models are a mathematical model that applies statistics and machine learning techniques to predict the performance of athletes in specific competitions. These prediction models can help coaches and athletes develop training plans, optimize training methods, and determine

**Data availability statement:** All relevant data are within the paper and its Supporting Information files.

**Funding:** The author(s) received no specific funding for this work.

**Competing interests:** N/A.

the potential and development direction of athletes. For example, the model for predicting the performance of track athletes proposed by Ref. [1], which uses exponential functions as fitting curves to predict the performance of track athletes. The model can be used to determine whether athletes are suitable for selection or whether training methods are effective and reasonable. Ref. [2] designed a model for predicting the performance of aerobics athletes in specialized sports. The model took these indicators including body shape, function, and fitness as independent variables, meanwhile, took sports performance as the dependent variable. This design greatly improves the accuracy of prediction results.

Unlike the design proposed by Ref. [2] and Ref. [3], Ref. [3] utilized the model-based BP neural networks to settle long jump performance prediction of athletes. The model can be used to analyze the technical characteristics and sports potential of long jump athletes, while assisting in guiding training and improving technical movements. Similarly, Ref. [4] took an artificial neural network model to predict the performance of shot-put athletes, through using an artificial neural network to learn the scores of shot-put athletes. Artificial neural network (ANN) models can learn and discover hidden patterns from a large amount of data, thereby accurately predicting the performance of athletes. ANN models are designed to better understand and predict the performance of athletes, thereby helping them achieve their optimal state. In practical applications, the parameters and structures of ANN models will be selected and adjusted according to different sports projects and specific needs to obtain the most suitable prediction effects.

### 1.1. Research motivation

This article aims to design a predictive model to analyze various factors that affect athlete training performance, to provide more scientific guidance for athlete training. Firstly, our primary goal is to achieve the forecast of training performance for football athletes. Then, the second goal is to find the major factors that have positive effects on the training performance. Accordingly, to finish the two goals, this paper took the Event-group training theory as the studied base, and constructed seven training indicators, i.e., strength training, speed training, endurance training, regional training, resisted stress training, recovery training, basic skill training. Following that, the discrete Hopfield neural networks (namely DHNNs) model is designed based on the seven training indicators. To improve the forecast ability of the DHHNNs model, a Markov probability transition matrix is used to calculate the activation probability of neurons. Finally, using the trained DHNNs model to forecast the training performance of football athletes.

### 1.2. Contributions

The contributions in this work are summarized as follows.

(1) We utilize the Event-group training theory to customize seven training indicators for football athletes. Meanwhile, using the designed model finds the major training indicators affecting the training performance in the seven training indicators. This provides scientific assistance for coaches to develop training plans of football athletes.

(2) The discrete Hopfield neural network fusing with Markov probability transition matrix is proposed. To accurately control the activation status of neurons, the Markov probability transition matrix is used to calculate the probability of neuron activation, thus improving the forecast performance of the proposed model.

(3) Since the Markov probability transition matrix is constructed based on the seven training indicators, the calculation time does not exhibit exponential running time. In terms of the

proposed model, the training time emerges a linear trend, therefore, the proposed model can be used for the forecast of large-scale datasets.

This paper is arranged as follows. The related works are reviewed in Section 2. Section 3 illustrates the proposed method, including the design of the seven training indicators and the development of the model. Experimental settings and experimental results are given in Section 4 and Section 5. Finally, Section 6 draws a conclusion and directs future work.

## 2. Related works

Machine learning methods can help coaches and athletes develop more scientific and personalized training plans through analyzing large amounts of sports data. For example, by analyzing players' movements and performances, it can provide targeted training suggestions to improve players' technical level and competitive ability. Sona et al [5] use a Random Forest model (RFM) to understand the health conditions of athletes and to predict the training performance. The experimental results prove the RF has good forecast accuracy on the test dataset. Zhu et al [6] develop a Bayesian model (BAM) fusing with the formalized description rules for movement to recognize incorrect movements in daily training, and to judge the training performance of athletes. Through applying the three-dimensional coordinates, the BAM obtains ideal recognition accuracy for incorrect movements, however, compared to the recognition accuracy, it has weak forecast ability in training performance. Gianluca et al [4] use a Multivariate Regression Model (MRM) to attempt forecasting football players' performance during training sessions, which makes tactical decisions and designs customized training sessions. Similarly, the machine learning approach implemented in Ref. [6–9] is also used for the forecast of training performance for athletes.

Fierce competitiveness has become one of the most essential characteristics that distinguishes competitive sports from other sports. Modern competitive sports require scientific training plans and standardized technical and tactical training [10,11]. Aiming at the characteristics of competitive sports, Nan [3] sufficiently applies artificial intelligence sensor to observe the training states of athletes. Then the observed training states are feedbacked to training plans, through correcting training plans, the training performance of athletes is improved. Like Ref. [3], Su et al [12] also apply artificial intelligence technology for physical Training. And the artificial intelligence technology in Ref. [13]. The smart Wireless Sensor Network (WSN) proposed in Ref. [14] focuses on training states of athletes. Through analyzing the training states, WSN can objectively evaluate training performance of athletes. WSN is an online monitoring approach, which has very high demand for real-time performance. Zheng [13] researches on the application of artificial intelligence technology in improving scientific training performance, and proposes that Artificial Intelligence Technology is more significant than machine learning methods in Improving Training Performance of athletes.

Neural network approaches exhibit a certain level of predictive ability in settling the forecast of training performance [15]. For example, the Leveraging tabular variational autoencoders (TVAE) [8] is used for the analytics of learning performance basketball players. The learning performance of TVAE are interpretable, and based on the interpretable results, training schemes can be recommended for optimal performance during the competition. Ping et al [16] take a Hopfield neural network (HMM) model with blockchain mechanism to predict sports performance. Although the designed structure of the HMM is simple, it operates efficiently and with high forecast accuracy due to taking blockchain mechanism. And Gui et al [17] borrow the forecast ability of Hopfield neural networks to achieve classifications, and suggest that Hopfield neural networks are suitable for the forecast.

## 3. Methodology

### 3.1. Overall scheme

The proposed scheme includes the design of seven training indicators, the implementation of the model, and training performance forecast. Firstly, according to Event-Group theory, seven training indicators affecting training performance for football athletes were designed. Then, the proposed model was illustrated. Thereafter, we designed the experimental dataset according to the seven training indicators. The experimental dataset was into the training set, the testing set, and the validation set. Using the training set to train the model, and the testing set is used for parameter validation of the model. Finally, using the validation set to verify the forecast ability of the model.

### 3.2. Designs of seven indicators

The Event-Group theory [2,18] is proposed by combining with the general training theory and specialized training theory of competitive sports. The Event-Group theory suggests that the basic tasks of training activities are to improve and develop the competitive abilities of athletes, and indicates that the determining factors of competitive ability include ideology, function, quality, and other aspects. Through analyzing these determining factors, coaches can gain a more comprehensive understanding to the various factors affecting competitive abilities of athletes.

The Event-Group theory involves the two aspects physical fitness category and skill category [2,18]. In the category of physical fitness, including strength, speed, and endurance. Whereas, in the skill category, including difficulty and beauties in performance, accuracy in performance, resistance across nets, resistance in the same field, and resistance in combat. The physical fitness category has a determination role, while the skill category plays a leading role. As such, using Event-Group theory to design seven training indicators, listed in Table 1. As follows,

Indicator 1, Strength Training (STT). Due to the decisive role of athletes' strength level in their competitive ability composition factors, the strength training is taken as a critical indicator. This work designed moderate intensity training and high-intensity training, where, the former requires 5 minutes in each training, and the latter needs 10 minutes in each training. With an hour, if a football athlete completed both 6-times of moderate intensity training and 3-times of high-intensity training, the athlete football obtains 10 points. If the football athlete only finishes one of the both trainings, 5 points can be obtained. Otherwise, the football athlete only receives 0 point.

Indicator 2, Speed Training (SPT). When a football athlete ran 100 meters running within certain time, the athlete receives 10 points, otherwise, the football athlete only obtains 0 point.

Indicator 3, Endurance Training (ENT). For the endurance indicator, aerobic training, anaerobic training, and hybrid training of the both are considered. Within an hour, when the

Table 1. Deals of the seven training indicators.

| Indicator 1 | Indicator 2 | Indicator 3 | Indicator 4 | Indicator 5 | Indicator 6 | Indicator 7 |
|---|---|---|---|---|---|---|
| STT | SPT | ENT | RET | RST | RCT | BST |
| Score settings | | | | | | |
| 10 points | 10 points | 10 points | 10 points | 10 points | 10 points | 10 points |
| 5 points | – | 5 points | 5 points | – | 5 points | – |
| 0 point | 0 point | 0 point | 0 point | 0 point | – | 0 point |

three trainings are finished together, the football athlete gains 10 points. If any two trainings are completed, the football athlete just gets 5 points. Otherwise, the football athlete obtains 0 point.

Indicator 4, Regional Training (RET). In the training, different training conditions were set up, including barrier-free training, obstacle training. Similarly, within an hour, if a football athlete can finish the both trainings, the football athlete can receive 10 points. If one of the both trainings was finished, 5 points can be obtained. Otherwise, the football athlete obtains 0 point.

Indicator 5, Resisted Stress Training (RST). This is to test football athlete's ability to withstand stress. After finishing high-intensity training, using a testing system to evaluate football athlete psychology. If the testing is passed, the football athlete receives 10 points, instead, there gets 0 point.

Indicator 6, Recovery Training (RCT), which is to observe the recovery of athletes. Using the designed recovery training plan to train football athletes, after finishing the recovery training, we utilized the monitoring equipment to record the recovery time that physiological indicators of athletes recover standard values, such as heart rate, blood glucose concentration, etc. If the recovery time of a football athlete is higher than average recovery time of the group, the football athlete only gets 5 points. Otherwise, the football athlete can obtain 10 points.

Indicator 7, Basic Skill Training (BST). Within an hour, if a football athlete can complete the basic training movements of football in a standard way, the football athlete can receive 10 points. Otherwise, 0 point is obtained.

## 3.3. Model implementation

**(1) Model structure.** The purpose is to predict the training performance of football athletes. Based on this, here, we took account into neural network architectures because of ascendency forecast ability. Additionally, for the seven discrete training indicators, we chose discrete Hopfield neural networks (DHNNs) that belong to recurrent neural networks.

Fig 1 displays the structure of DHNNs. Each neuron plays the same role, that is, neuron $x_i$ and neuron $x_j$ have the same state. The state $X = [x_1, x_2, .., x_n]^{\mathrm{T}}$ of DHNNs consists of the set of neuron state, and the initialization state is denoted as $X(0) = [x_1(0), x_2(0), .., x_n(0)]^{\mathrm{T}}$. The state is changed by Eq. (1).

$$x_j = f(\mathrm{net}_j) \qquad j = 1,2,...,n \tag{1}$$

where $f(\bullet)$ is the transformation function, as follows,

$$x_j = \mathrm{sgn}(\mathrm{net}_j) = \begin{cases} 1, & \mathrm{net}_j \geq 0 \\ -1, & \mathrm{net}_j < 0 \end{cases} \qquad j = 1,2,...,n \tag{2}$$

sgn is a symbol function. Having that

$$\mathrm{net}_j = \sum_{i=1}^{n}(w_{ij}x_i - T_j) \quad j = 1,2,...,n \tag{3}$$

$w_{ij}$ is a weight value, and $w_{ij} = w_{ji}$, $w_{ij} = 0$. $T_j$ is a threshold. When the state of each neuron is not changed, the output state of the network is as follows,

$$\lim_{t \to \infty} x(t) \tag{4}$$

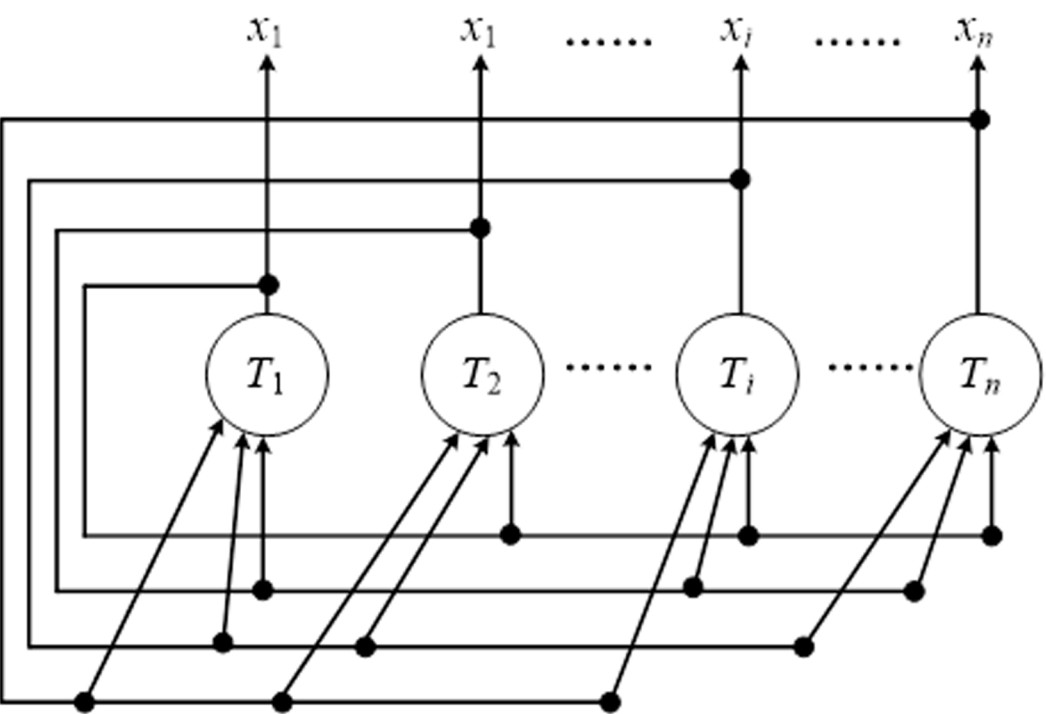

**Fig 1. The structure of DHNNs.**

**(2) Calculation of threshold $T_j$.** We take the Markov probability transition matrix to calculate the threshold $T_j$ in Eq. (3). Assuming that there are $m$ neurons, we initialize the first-step Markov probability transition matrix $P(1)$.

$$P(1) = \begin{bmatrix} P_{11} & P_{12} & ... & P_{1m} \\ P_{21} & P_{22} & ... & P_{2m} \\ ... & ... & ... & ... \\ P_{m1} & P_{m2} & ... & P_{mm} \end{bmatrix} \tag{5}$$

Here, we discuss the calculation of $P(1)$. For the convenience of illustrate, assuming that

the training dataset is denoted as $STrain = \begin{bmatrix} S_{11} & S_{12} & ... & S_{1K} \\ S_{21} & S_{22} & ... & S_{2K} \\ ... & ... & ... & ... \\ S_{V1} & S_{V2} & ... & S_{VK} \end{bmatrix}$, where $K$ is the data dimen-

sion and $V$ is the data volume. There are two situations for the training dataset $STrain$. Among them, one is that the $K$ are equal to $V$, and the other is that they are not equal. The two situations are discussed, respectively.

**Situation (*i*)**, when $K == V$, i.e., the data dimension is equal to the data volume. In this

situation, the training set $STrain = \begin{bmatrix} S_{11} & S_{12} & ... & S_{1K} \\ S_{21} & S_{22} & ... & S_{2K} \\ ... & ... & ... & ... \\ S_{K1} & S_{K2} & ... & S_{KK} \end{bmatrix}$. Let us consider the component in the

*i*-th row of *STrain*, i.e., $S_i^{'} = [S_{i1} \ S_{i2} \ ... \ S_{iK}]$, and $sum(S_i^{'}) = S_{i1} + S_{i2} + ... + S_{ik}$, then using Eq. (6) to handle the component $S_i^{'}$.

$$S_i^{'} = [\frac{S_{i1}}{sum(S_i^{'})} \ \frac{S_{i2}}{sum(S_i^{'})} \ ... \ \frac{S_{iK}}{sum(S_i^{'})}] \tag{6}$$

Each row component of the training set *STtrain* is handled by using Eq. (6). As follows,

$$STrain = \begin{bmatrix} \dfrac{S_{11}}{sum(S_1^{'})} & \dfrac{S_{12}}{sum(S_1^{'})} & ... & \dfrac{S_{1K}}{sum(S_1^{'})} \\ \dfrac{S_{21}}{sum(S_2^{'})} & \dfrac{S_{22}}{sum(S_2^{'})} & ... & \dfrac{S_{2K}}{sum(S_2^{'})} \\ ... & ... & ... & ... \\ \dfrac{S_{i1}}{sum(S_i^{'})} & \dfrac{S_{i2}}{sum(S_i^{'})} & ... & \dfrac{S_{iK}}{sum(S_i^{'})} \\ ... & ... & ... & ... \\ \dfrac{S_{K1}}{sum(S_K^{'})} & \dfrac{S_{K2}}{sum(S_K^{'})} & ... & \dfrac{S_{KK}}{sum(S_K^{'})} \end{bmatrix} \tag{7}$$

$P(1)$ is calculated by Eq. (7), i.e., $P_{11} = \frac{S_{11}}{sum(S_1^{'})}, ..., P_{1m} = \frac{S_{1K}}{sum(S_1^{'})}, ..., P_{m1} = \frac{S_{K1}}{sum(S_K^{'})}, ..., P_{mm} = \frac{S_{KK}}{sum(S_K^{'})}$. Having

$$P(1) = \begin{bmatrix} P_{11} & P_{12} & ... & P_{1m} \\ P_{21} & P_{22} & ... & P_{2m} \\ ... & ... & ... & ... \\ P_{m1} & P_{m2} & ... & P_{mm} \end{bmatrix} = \begin{bmatrix} \dfrac{S_{11}}{sum(S_1^{'})} & \dfrac{S_{12}}{sum(S_1^{'})} & ... & \dfrac{S_{1K}}{sum(S_1^{'})} \\ \dfrac{S_{21}}{sum(S_2^{'})} & \dfrac{S_{22}}{sum(S_2^{'})} & ... & \dfrac{S_{2K}}{sum(S_2^{'})} \\ ... & ... & ... & ... \\ \dfrac{S_{i1}}{sum(S_i^{'})} & \dfrac{S_{i2}}{sum(S_i^{'})} & ... & \dfrac{S_{iK}}{sum(S_i^{'})} \\ ... & ... & ... & ... \\ \dfrac{S_{K1}}{sum(S_K^{'})} & \dfrac{S_{K2}}{sum(S_K^{'})} & ... & \dfrac{S_{KK}}{sum(S_K^{'})} \end{bmatrix} \tag{8}$$

where $m = K$, i.e., the number of neurons is equal to the data dimension of training set *STrain*.

**Situation (ii)**, when $K \neq V$, i.e., the data dimension is not equal to the data volume. In this situation, we calculate the transpose matrix of $STrain$, i.e., $STrain^T = \begin{bmatrix} S_{11} & S_{12} & ... & S_{1K} \\ S_{21} & S_{22} & ... & S_{2K} \\ ... & ... & ... & ... \\ S_{V1} & S_{V2} & ... & S_{VK} \end{bmatrix}^T$

. If the data dimension is greater than the data volume, i.e., $K > V$, using Eq. (9) calculates $STrain^T$ and $STrain$.

$$STrain_{(K,V)} = STrain^T \times STrain = \begin{bmatrix} S_{11} & S_{12} & ... & S_{1K} \\ S_{21} & S_{22} & ... & S_{2K} \\ ... & ... & ... & ... \\ S_{V1} & S_{V2} & ... & S_{VK} \end{bmatrix}^T \begin{bmatrix} S_{11} & S_{12} & ... & S_{1K} \\ S_{21} & S_{22} & ... & S_{2K} \\ ... & ... & ... & ... \\ S_{V1} & S_{V2} & ... & S_{VK} \end{bmatrix} = \begin{bmatrix} S_{11} & S_{12} & ... & S_{1K} \\ S_{21} & S_{22} & ... & S_{2K} \\ ... & ... & ... & ... \\ S_{K1} & S_{K2} & ... & S_{KK} \end{bmatrix} \quad (9)$$

Clearly, $STrain_{(K,V)}$ is a matrix of *K* rows and *K* columns. Similarly, using this manner of Eq. (7) to calculate each row component of Eq. (9). Therefore, Eq. (9) is transformed into Eq. (10).

$$STrain_{(K,V)} = \begin{bmatrix} \frac{S_{11}}{sum(S_1^{'})} & \frac{S_{12}}{sum(S_1^{'})} & ... & \frac{S_{1K}}{sum(S_1^{'})} \\ \frac{S_{21}}{sum(S_2^{'})} & \frac{S_{22}}{sum(S_2^{'})} & ... & \frac{S_{2K}}{sum(S_2^{'})} \\ ... & ... & ... & ... \\ \frac{S_{i1}}{sum(S_i^{'})} & \frac{S_{i2}}{sum(S_i^{'})} & ... & \frac{S_{iK}}{sum(S_i^{'})} \\ ... & ... & ... & ... \\ \frac{S_{K1}}{sum(S_K^{'})} & \frac{S_{K2}}{sum(S_K^{'})} & ... & \frac{S_{KK}}{sum(S_K^{'})} \end{bmatrix} \quad (10)$$

Utilizing Eq. (10) to calculating $P(1)$. As follows,

$$P(1) = \begin{bmatrix} P_{11} & P_{12} & ... & P_{1m} \\ P_{21} & P_{22} & ... & P_{2m} \\ ... & ... & ... & ... \\ P_{m1} & P_{m2} & ... & P_{mm} \end{bmatrix} = \begin{bmatrix} \frac{S_{11}}{sum(S_1^{'})} & \frac{S_{12}}{sum(S_1^{'})} & ... & \frac{S_{1K}}{sum(S_1^{'})} \\ \frac{S_{21}}{sum(S_2^{'})} & \frac{S_{22}}{sum(S_2^{'})} & ... & \frac{S_{2K}}{sum(S_2^{'})} \\ ... & ... & ... & ... \\ \frac{S_{i1}}{sum(S_i^{'})} & \frac{S_{i2}}{sum(S_i^{'})} & ... & \frac{S_{iK}}{sum(S_i^{'})} \\ ... & ... & ... & ... \\ \frac{S_{K1}}{sum(S_K^{'})} & \frac{S_{K2}}{sum(S_K^{'})} & ... & \frac{S_{KK}}{sum(S_K^{'})} \end{bmatrix} \quad (11)$$

where $m = K$, that is, the number of neurons is equal to the data dimension of training set *STrain*.

If $K < V$ holds, this means that the data dimension is smaller than the data volume, then the $STrain^{\mathrm{T}}$ and $STrain$ are calculated by Eq. (12).

$$STrain_{(V,K)} = STrain \times STrain^{\mathrm{T}} = \begin{bmatrix} S_{11} & S_{12} & \cdots & S_{1K} \\ S_{21} & S_{22} & \cdots & S_{2K} \\ \cdots & \cdots & \cdots & \cdots \\ S_{V1} & S_{V2} & \cdots & S_{VK} \end{bmatrix} \times \begin{bmatrix} S_{11} & S_{12} & \cdots & S_{1K} \\ S_{21} & S_{22} & \cdots & S_{2K} \\ \cdots & \cdots & \cdots & \cdots \\ S_{V1} & S_{V2} & \cdots & S_{VK} \end{bmatrix}^{\mathrm{T}} = \begin{bmatrix} S_{11} & S_{12} & \cdots & S_{1V} \\ S_{21} & S_{22} & \cdots & S_{2V} \\ \cdots & \cdots & \cdots & \cdots \\ S_{V1} & S_{V2} & \cdots & S_{VV} \end{bmatrix}$$

(12)

$STrain_{(V,K)}$ is a matrix of $V$ rows and $V$ columns. Similarly, using this manner of Eq. (7) to calculate each row component of Eq. (12). As such, Eq. (12) can be transformed into Eq. (13).

$$STrain_{(V,K)} = \begin{bmatrix} \dfrac{S_{11}}{sum(S_1')} & \dfrac{S_{12}}{sum(S_1')} & \cdots & \dfrac{S_{1V}}{sum(S_1')} \\ \dfrac{S_{21}}{sum(S_2')} & \dfrac{S_{22}}{sum(S_2')} & \cdots & \dfrac{S_{2V}}{sum(S_2')} \\ \cdots & \cdots & \cdots & \cdots \\ \dfrac{S_{i1}}{sum(S_i')} & \dfrac{S_{i2}}{sum(S_i')} & \cdots & \dfrac{S_{iV}}{sum(S_i')} \\ \cdots & \cdots & \cdots & \cdots \\ \dfrac{S_{V1}}{sum(S_V')} & \dfrac{S_{V2}}{sum(S_V')} & \cdots & \dfrac{S_{VK}}{sum(S_V')} \end{bmatrix}$$

(13)

Utilizing Eq. (13) to calculating $P(1)$, having that

$$P(1) = \begin{bmatrix} P_{11} & P_{12} & \cdots & P_{1m} \\ P_{21} & P_{22} & \cdots & P_{2m} \\ \cdots & \cdots & \cdots & \cdots \\ P_{m1} & P_{m2} & \cdots & P_{mm} \end{bmatrix} = \begin{bmatrix} \dfrac{S_{11}}{sum(S_1')} & \dfrac{S_{12}}{sum(S_1')} & \cdots & \dfrac{S_{1V}}{sum(S_1')} \\ \dfrac{S_{21}}{sum(S_2')} & \dfrac{S_{22}}{sum(S_2')} & \cdots & \dfrac{S_{2V}}{sum(S_2')} \\ \cdots & \cdots & \cdots & \cdots \\ \dfrac{S_{i1}}{sum(S_i')} & \dfrac{S_{i2}}{sum(S_i')} & \cdots & \dfrac{S_{iV}}{sum(S_i')} \\ \cdots & \cdots & \cdots & \cdots \\ \dfrac{S_{V1}}{sum(S_V')} & \dfrac{S_{V2}}{sum(S_V')} & \cdots & \dfrac{S_{VK}}{sum(S_V')} \end{bmatrix}$$

(14)

where $m = V$, that is, the number of neurons is equal to the data volume of training set $STrain$.

Based on the above analysis, the first-step Markov probability transition matrix $P(1)$ can be calculated by Eq. (16)

$$P(1) = \begin{cases} \text{Eq.(8)} \;\; and \; m = K & \text{if } K == V \\ \text{Eq.(11)} \; and \; m = K & \text{if } K > V \\ \text{Eq.(14)} \; and \; m = V & \text{if } K < V \end{cases}$$

(15)

where $K$, $V$ are the data dimension and data volume of training set $STrain$, respectively. Item $m$ is the number of neurons. The $m$-step Markov probability transition matrix $P(m)$ can be calculated by $P(1)$.

$$P(m) = P(m-1)*P(m-2)*....*P(1) = \begin{bmatrix} P_{11}^m & P_{12}^m & ... & P_{1m}^m \\ P_{21}^m & P_{22}^m & ... & P_{2m}^m \\ ... & ... & ... & ... \\ P_{m1}^m & P_{m2}^m & ... & P_{mm}^m \end{bmatrix} \tag{16}$$

For the component in the $i$-$th$ row of $P(m)$, denoted as $P(m)_i^{'} = [P_{i1}^m \; P_{i2}^m \; ... \; P_{im}^m]$, for the threshold $T_i$ corresponding to the $i$-$th$ neuron, we chose the maximum value in the $P(m)_i^{'}$ for $T_i$, illustrated in Eq. (17).

$$T_i = \max \; [P_{i1}^m \; P_{i2}^m \; ... \; P_{im}^m] \tag{17}$$

Eq. (17) indicates that the DHNNs configurates $m$ neurons, and the $i$-$th$ neuron is activated with a probability of $T_i$. Choosing the maximum value in the $P(m)_i^{'}$ is also to maximize the probability that the neuron is activated.

**(3) Model training.** The training of the DHNNs is illustrated in Algorithm 1. The input is datasets $S$, and the output is the forecast accuracy. The parameters are initialized in Step 1, and the datasets $S$ is divided into the training set $STtrain$, the testing set $STest$ and the validation set $SValidation$ in Step 2 to Step 4. Following that, the $m$-step Markov probability transition

**Algorithm 1. Training of DHNNs.**

| | |
|---|---|
| Input: Dataset S. | |
| Output: training accuracy, predicted accuracy. | |
| 1 | Initializing parameters; |
| 2 | 60% of S is used for STrain;  /* training set */ |
| 3 | 20% of S is used for STest;   /* testing set */ |
| 4 | 20% of S is used for SValidation;  /* verifying set */ |
| 5 | Using Eq. (16) to calculate $P(m)$; |
| 6 | Using Eq. (17) to calculate $T_i$; |
| 7 | **For** j = 1 **to** Jmax **do**: |
| 8 | STrain trains DHNNs; |
| 9 | Using Eq. (3) to update network states; |
| 10 | **if** DHNNs(STrain, j) convergence is true **then**: |
| 11 | Obtaining the training accuracy Accuracy_Training; |
| 12 | **break;** |
| 13 | **end if** |
| 14 | **end for** |
| 15 | Saving current training parameters for DHNNs(STrain); |
| 16 | Using STest to test the trained DHNNs(STest); |
| 17 | Outputting the testing accuracy; |
| 18 | Using SValidation to verify DHNNs(SValidation); |
| 19 | Outputting the predicted accuracy; |

matrix $P(m)$ can be calculated by [Eq. (16)](), and the threshold $T_i$ is calculated by [Eq. (17)](), illustrated in Step 5 and Step 6. The DNHHs is iteratively trained $J_{max}$-times, as shown in the procedure between Step 7 to Step 14. Using [Eq. (3)]() to update network states, until the DHNNs can converge, the training is terminated and the current training parameters are saved, then the training accuracy is outputted. In Step 16 and Step 17, the testing set *STest* is used for parameter testing of DHNNs. Finally, using the validation set *SValidation* to verify DHNN, and the forecast accuracy is outputted in Step 18 and Step 19.

## 4. Experimental settings

### 4.1. Datasets and evaluated indicators

We chose 20 football athletes to train 30 days, and collected the data. Each athlete needs to train 4 times per day, according to [Table 1](), each athlete can receive corresponding the points for completing a training indicator. Accordingly, the experimental dataset consists of 2400 rows and 7 columns, illustrated in [Table 2](). This meaning that the experimental dataset contains 16800 data.

The two metrics Precision and F1-score are used to evaluate the forecast ability of our DHNNs and the opponents. Apart from our DHNNs, we chose four competitors Random Forest model (RFM) [5], Bayesian model (BAM) [6], Multivariate Regression model (MRM) [4] and Tabular Variational Autoencoders (TVAE) [8] to compare against our DHNNs.

### 4.2. Experimental designs

The followed experiments were designed to test our model. To have a fair comparison, using the same training set *STrain* to the four opponents RFM, BAM, MRM and TVAE, and using the same testing set *STest* to test their parameters. Finally, once the four opponents and our DHNNs are trained well, using the same validation set *SValidation* to verify them.

Experiment (I). Indicator analytics. Through calculating the correlation between the forecast training performance of DHNNs and the seven training indicators in Tabel 2, then, the major training indicators affecting the training performance of football athletes are discussed.

Experiment (II). Comparisons of forecast ability. Through using the validation set *SValidation* to compare DHNNs with the four opponents RFM, BAM, MRM and TVAE, then, compared results are discussed.

Experiment (III). Executing efficiency. We compared the training time of DHNNs with the four opponents through using the training set *STrain*. Then, their running time is discussed.

**Table 2. Dataset details.**

| ID | Training frequency | Indicator 1 | Indicator 2 | Indicator 3 | Indicator 4 | Indicator 5 | Indicator 6 | Indicator 7 |
|---|---|---|---|---|---|---|---|---|
| A1 | 1st | 10 | 10 | 10 | 10 | 10 | 10 | 10 |
| A1 | 2se | 5 | 0 | 5 | 5 | 0 | 5 | 10 |
| A1 | 3rd | 0 | 0 | 10 | 0 | 0 | 10 | 0 |
| A1 | 4th | 10 | 0 | 5 | 0 | 10 | 10 | 10 |
| A2 | 1st | 5 | 10 | 10 | 5 | 10 | 10 | 10 |
| … | … | … | … | … | … | … | … | … |
| A20 | 4th | 10 | 10 | 10 | 0 | 10 | 10 | 10 |

## 5. Results analysis

The experimental results show that the proposed DHNNs wins over the four opponents RFM, BAM, MRM and TVAE in forecast results, and outperforms the opponent TVAE in running efficiency. The details are as follow.

### 5.1. Indicator analytics

To sort the seven training indicators, the correlation coefficient of the forecast training performance obtained by DHNNs and the seven training indicators are calculated, as shown in Fig 2. It can be seen that the three training indicators BST, SPT and STT have a strong correlation with the forecast training performance. However, the two training indicators RET and RCT show a weak correlation with the forecast training performance. Additionally, we used the SPSS tool to analyze the relations between the seven training indicators and the forecast training performance, illustrated in Table 3. Analyzed results show that the three training indicators BST, SPT and STT still exist a strong correlation with the forecast training performance.

Together, these results in Fig 2 and Table 3 confirm that the improvement of the training performance for football athletes depends more on basic skill training (BST), speed training (SPT) and strength training (STT), whereas, regional training (RET) and recovery training (RCT) provide less assistance in improving the training performance for football athletes. As such, we demonstrate that football athletes need to strengthen basic skill training, speed training and strength training to quickly improve the training performance.

To further observe the seven training indicators, we used $k$-means clustering to cluster the seven training indicators, as shown in Fig 3. Where the number of the clustering is equal to 3, i.e., $k = 3$. Through observing the Fig 3, the clustering effectiveness of the seven training

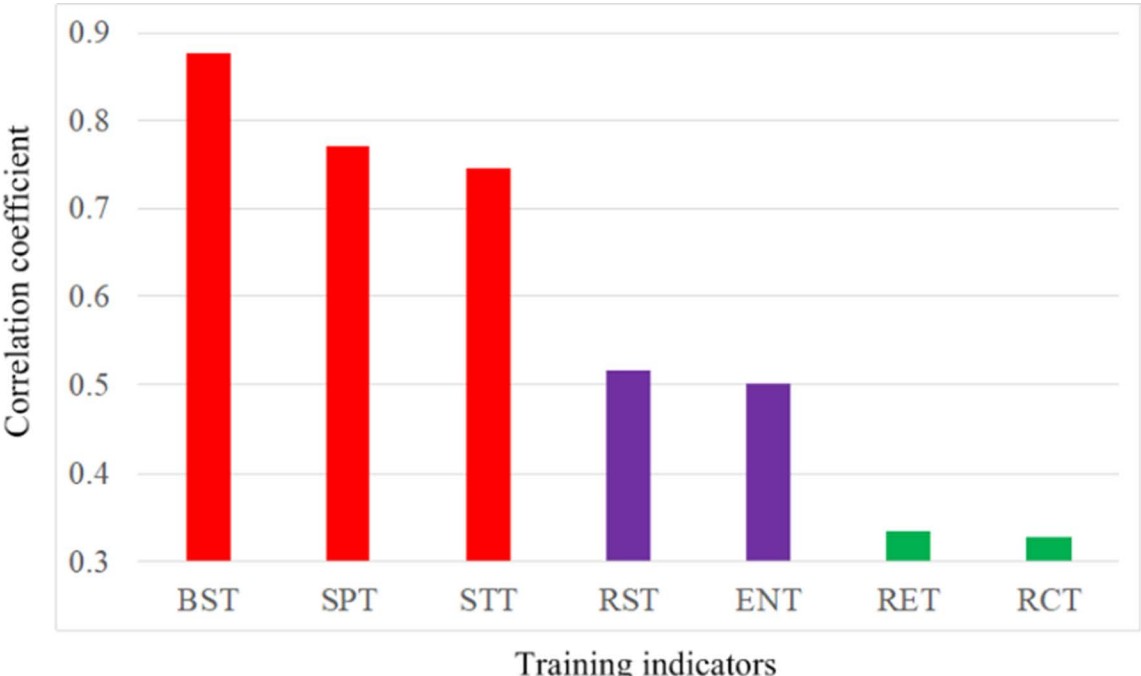

**Fig 2. Correlation analytics.** Correlation coefficient between the seven training indicators and the forecast training performance.

Table 3. Correlations of seven training indicators and forecast training performance. Sign ** means that there is a significant correlation at 0.05 level (two tailed).

| Training indicators | Pearson correlation | Training performance *Sig(two tailed)* | Number of training samples |
|---|---|---|---|
| RET | 0.336 | 0.346 | 1440 |
| RCT | 0.329 | 0.322 | 1440 |
| RST | 0.522 | 0.440 | 1440 |
| ENT | 0.497 | 0.432 | 1440 |
| BST | **0.881 ** | 0.000 | 1440 |
| SPT | **0.779 ** | 0.000 | 1440 |
| STT | **0.748 ** | 0.000 | 1440 |

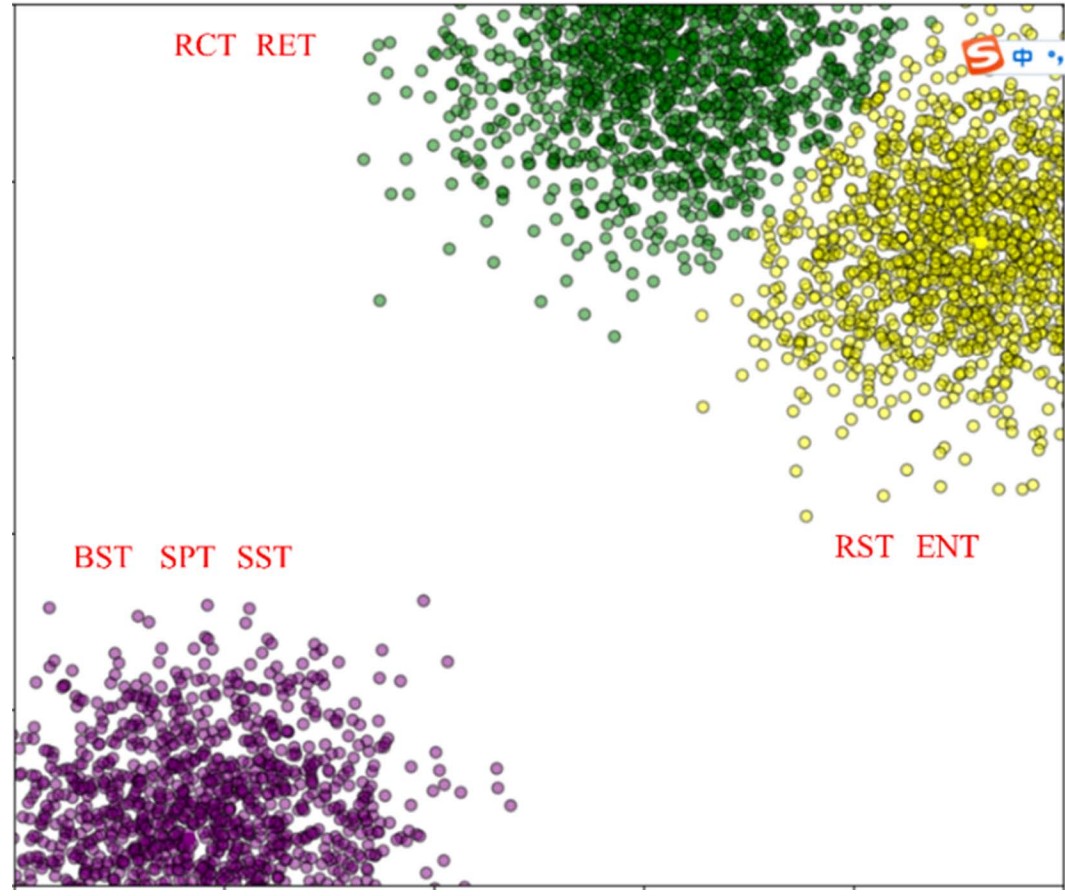

**Fig 3. Clustering for the seven training indicators.** The number of clustering is 3. The three indicators BST, SPT and SST are marked with purple circles. The two indicators RST and ENT are marked with yellow circles. The two Indicators RCT and RET are marked with green circles.

indicators are significant from the perspective of correlations when $k = 3$ holds. Consequently, these clustered results in Fig 3 are consistent with those in Fig 2 and Table 3. In summary, it is recommended to choose the three training indicators BST, SPT, STT as the critical training indicators to quickly improve the training performance of football athletes.

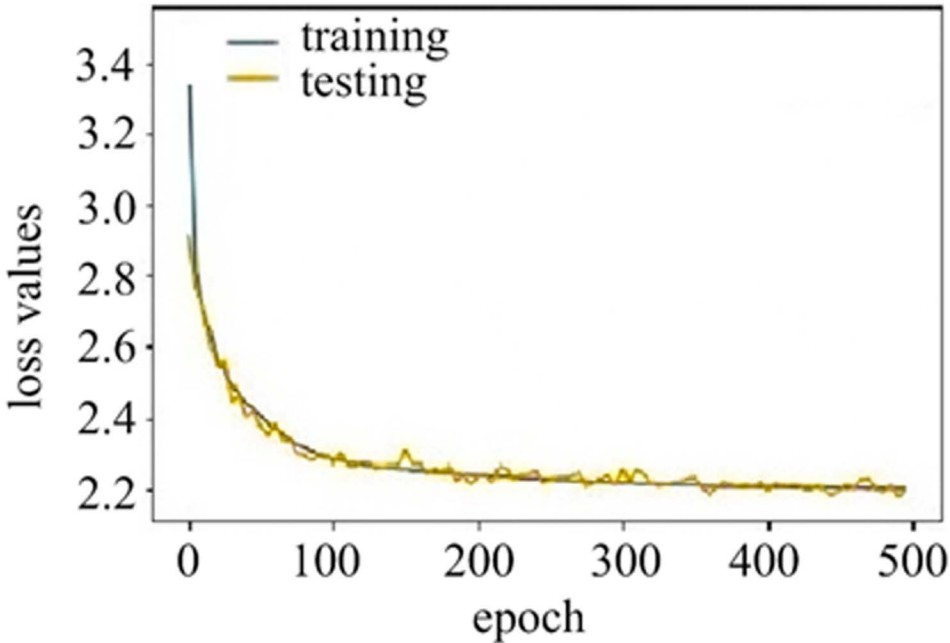

**Fig 4.** Loss values. Yellow curve is loss value of the testing. Green curve is loss value of the training.

For our DHNNs, the training loss and testing loss are given in Fig. 4. It can be seen that DHNNs begins to converge at 380 epoch. Furthermore, we can observe that there is no significant oscillation in the loss value curves in the process of DHNNs' training. Therefore, our DHNNs did not exhibit over-fitting during training.

## 5.2. Comparisons of forecast ability

In this section, the forecast ability of our DHNNs and that of the four opponents RFM, BAM, MRM and TVAE are illustrated in Fig 5 and Table 4. Through using the four-evaluation metrics Accuracy, Precision, G-score and F1-score, our DHNNs wins over the four opponents, implying that our DHNNs has ability to forecast the training performance for football athletes. In terms of the four competitors, the competitor BAM has the weakest predictive ability, however, the competitor TVAE outperforms the three competitors RFM, BAM, MRM. Overall, the two models DHNNs and TVAE based on neural network structures have more advantages than the three models RFM, BAM, MRM in the prediction of training performance for football athletes.

Furthermore, from Table 4, it can be seen that our DHNNS is better than the four competitors at 95% confidence level. These statistical results show that there is no difference between our DHNNs and the four opponents TVAE, RFM, BAM and MRM.

## 5.3. Executing efficiency

Fig 6 displays the training time of the five models. The opponent RF obtains the lowest training time, and the opponent TVAE obtains the highest training time. However, the training time of our DHNNs only defeats against the opponent TVAE. For the training set *STrain*, the data volume and the data dimension are $V$ and $K$, respectively, as for the time complexity of our DHNNs, the time consumption contains the calculation time $O(V * K)$ of the

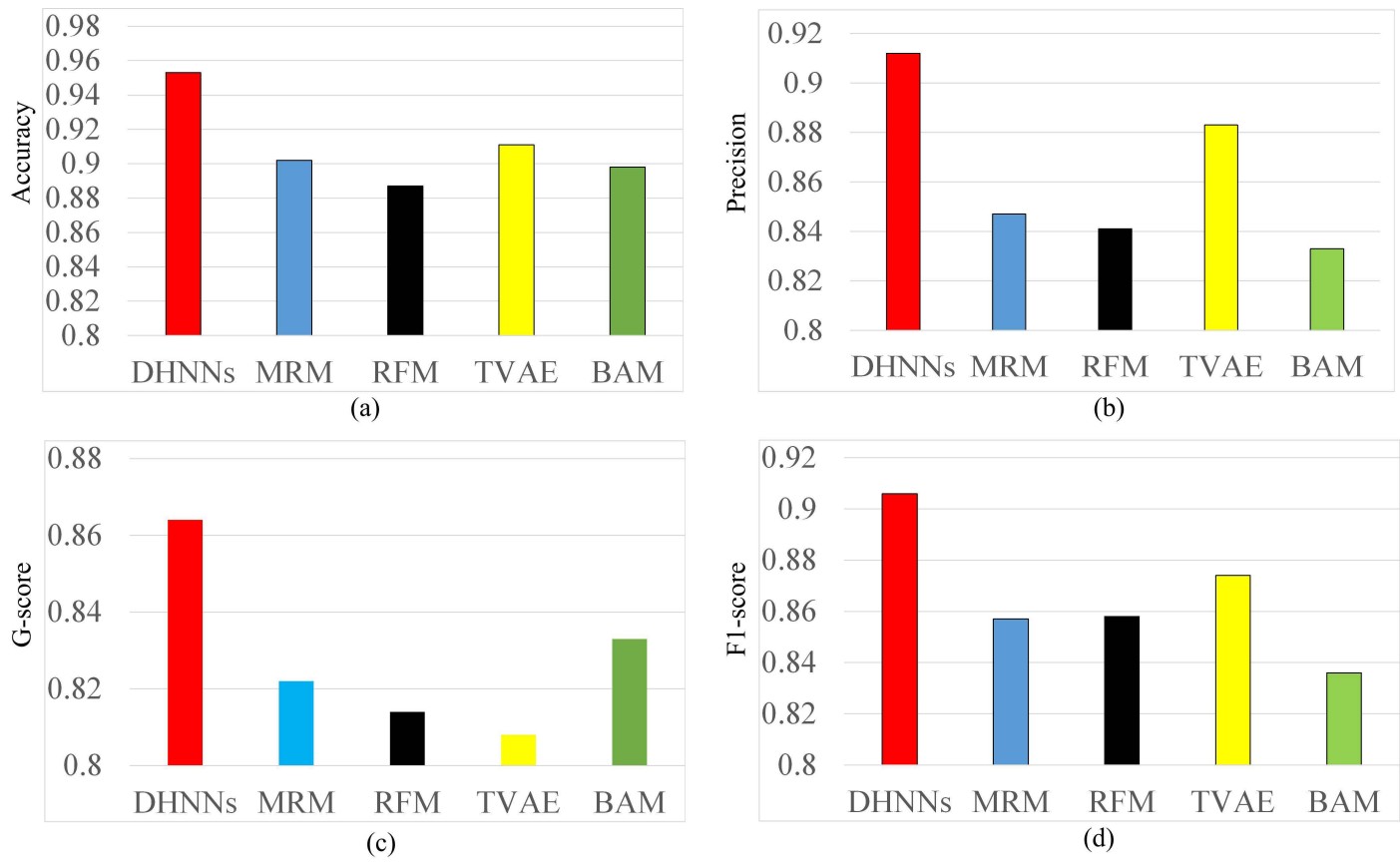

**Fig 5. Forecast performance of the five models.**

**Table 4.** Forecast results. *p*-value is given in the last row, and the sign '*' shows significant at *p* = 0.05 level.

| Methods | DHNNs | MRM | RFM | TVAE | BAM |
|---|---|---|---|---|---|
| metrics Accuracy, [Precision], {F1-score}, <G-score> | | | | | |
| Validation dataset | 0.953 | 0.902 | 0.887 | 0.911 | 0.898 |
| | [0.912] | [0.847] | [0.841] | [0.883] | [0.832] |
| | {0.906} | {0.857} | {0.858} | {0.874} | {0.836} |
| | <0.864> | <0.822> | <0.815> | <0.808> | <0.833> |
| *p* = 0.05 | * | * | * | * | * |

$m$-step Markov probability transition matrix $P(m)$ and the update time $O(J_{max})$ of network status. (Since DHNNs is trained $J_{max}$-times). Therefore, the time complexity of DHNNs is $O(\text{DHNNs}) = O(V * K) + O(J_{max})$. The training time of the competitor RFM spends the traversing of tree depth, hence, the time complexity of BFM is $O(\text{RFM}) = O(V * D_{tree})$, where $D_{tree}$ is the tree depth. While for the two competitors BAM and MRM, the time complexity is $O(\text{BAM}) = O(V * K)$ and $O(\text{MRM}) = O(V * K)$, respectively. While for the competitor TVAE, the time spend data processing, therefore, the time complexity is $O(\text{TVAE}) > O(V * K)$.

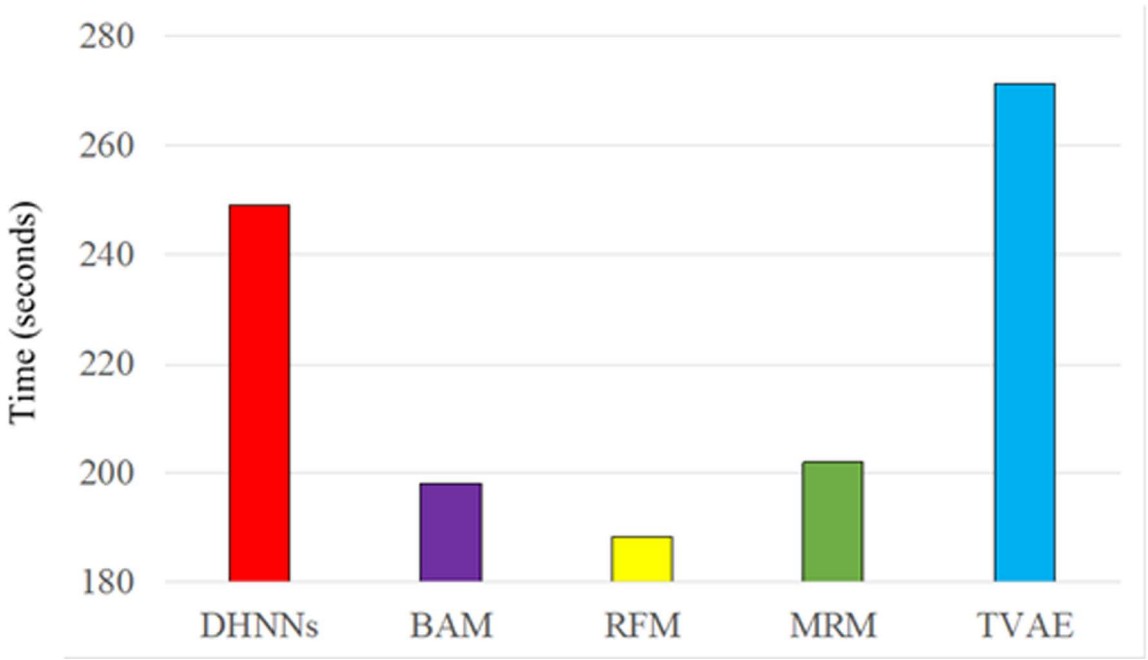

**Fig 6. Training time of the five methods.**

## 6. Conclusion

To predict the training performance of football athletes, this paper proposed a neural network based on the Markov probability transition matrix. The principle is that the Event-group training theory is used to design seven training indicators, thereafter, a discrete Hopfield neural network is developed based on the seven training indicators. Since the states of neurons affects the forecast ability of the discrete Hopfield neural network, this work took account into Markov probability transition matrix to control the activation probability of neurons. By doing so, the proposed model can effectively predict the training performance. Experimental results show that the proposed model wins over the comparative methods in the forecast of training performance of football athletes. And results also imply that the seven training indicators can provide scientific assistance for coaches to customize training plans. Moreover, the seven training indicators have ability to evaluate the effectiveness of training plans in promoting training performance for football athletes. In future work, we will put the seven training indicators into other competitive sports, and will verify their effectiveness from different perspectives. Then, based on the seven training indicators, we will expand more training indicators affecting the training performance of athletes.

## Supporting information

**S1 Data. Final source dataset.**
(TXT)

## Author contributions

**Conceptualization:** hongxing peng.

**Data curation:** Li Li, Long Cheng.

**Investigation:** hongxing peng.

**Methodology:** hongxing peng.

**Software:** hongxing peng, Li Li, Long Cheng.

**Validation:** hongxing peng, Li Li, Long Cheng.

**Writing – original draft:** hongxing peng.

**Writing – review & editing:** hongxing peng.

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
