## [Editor Report · Decision Letter 0]

25 Sep 2024

Dear Dr. peng,

Thank you for submitting your manuscript to PLOS ONE. After careful consideration, we feel that it has merit but does not fully meet PLOS ONE’s publication criteria as it currently stands. Therefore, we invite you to submit a revised version of the manuscript that addresses the points raised during the review process.

We look forward to receiving your revised manuscript.

Kind regards,

Perepi Rajarajeswari

Academic Editor

PLOS ONE

Additional Editor Comments:

Dear sir,

Add more new contributions to the manuscript

Apply Evaluation metric and testing strategies to the proposed model

Technically paper is sound

Use technical writing for improving the quality of manuscript.

The manuscript will be accepted after making changes.

---

## [Author Response · Author response to Decision Letter 0]

1 Oct 2024

Dear Editor and Reviewers,

Many thanks for your insightful and informative comments which are very helpful for the further improvement of this work. We have revised this manuscript according to the reviewers’ comments and gave the detailed responses to their questions one by one in the following “Response to Reviewers’ comments”.

Best regards,

Hongxing Peng

Comments:

1. Add more new contributions to the manuscript

Response:

Many thanks for the insight comments. The revision was marked with the red.

New contributions are added, and please see Line #78 to Line #81.

2. Apply Evaluation metric and testing strategies to the proposed model

Response:

We applied two evaluation metrics Precision and F1-score to test the forecast ability of the proposed model, please see Section 5.2 in Line # 295 to Line #304. Additionally, we compared the running efficiency of the proposed model with that of the competitors. Please see Section 5.3 in Line #305 to Line #318.

3. Technically paper is sound.

Response:

Many thanks for the comments.

4. Use technical writing for improving the quality of manuscript.

Response:

We revised carefully this paper and made some minor adjustments to highlight the contributions.

5. The manuscript will be accepted after making changes.

Response:

Many thanks for the comments

---

## [Decision Letter · Decision Letter 1]

12 Nov 2024

Dear Dr. peng,

Thank you for submitting your manuscript to PLOS ONE. After careful consideration, we feel that it has merit but does not fully meet PLOS ONE’s publication criteria as it currently stands. Therefore, we invite you to submit a revised version of the manuscript that addresses the points raised during the review process.

We look forward to receiving your revised manuscript.

Kind regards,

Perepi Rajarajeswari

Academic Editor

PLOS ONE

Additional Editor Comments:

Dear sir,

Pls do all the comments according to reviewer comments and follow journal guidelines.

With regards

Dr Perepi Rajarajeswari

Academic editor

Reviewers' comments:

Reviewer's Responses to Questions

**Comments to the Author**

Reviewer #1: (No Response)

2. Is the manuscript technically sound, and do the data support the conclusions?

Reviewer #1: No

3. Has the statistical analysis been performed appropriately and rigorously?

Reviewer #1: No

4. Have the authors made all data underlying the findings in their manuscript fully available?

Reviewer #1: Yes

5. Is the manuscript presented in an intelligible fashion and written in standard English?

Reviewer #1: No

Reviewer #1: This paper proposes a neural network based on the Markov probability transition matrix to predict the training performance of football athletes. This paper has been revised according to the previous comments. However, the readability and writing are unsatisfactory.

Some comments are as follows:

1. The language should be polished. The logic in Section 1 Introduction is confused.

2. The quality of Figure 1 “The structure of DHNNs.” is unsatisfactory.

3. The control scheme seem to be simple validate of the existing predictive algorithm. The highlights of the work is unobvious.

4. The simulation process is too simple. Only bar charts are provided cannot be accepted.

**Do you want your identity to be public for this peer review?** For information about this choice, including consent withdrawal, please see our Privacy Policy

Reviewer #1: **Yes: ** Xin Hu

---

## [Author Response · Author response to Decision Letter 1]

25 Dec 2024

Dear Editor and Reviewers,

Many thanks for your insightful and informative comments. We have revised this manuscript according to the reviewers’ comments and gave the detailed responses to their questions one by one in the following “Response to Reviewers’ comments”.

Best regards,

Yours sincerely,

Hongxing Peng

Reviewer #1: This paper proposes a neural network based on the Markov probability transition matrix to predict the training performance of football athletes. This paper has been revised according to the previous comments. However, the readability and writing are unsatisfactory.

Some comments are as follows:

1. The language should be polished. The logic in Section 1 Introduction is confused.

Many thanks for the insightful comments. The revision was marked with the red.

Response:

We rewrote the Section 1 Introduction and organized the section to understand the logical. Please see Line #30 to Line #92.

2. The quality of Figure 1 “The structure of DHNNs.” is unsatisfactory.

Response:

We re-drew the Fig. 1, please see Line #199.

3. The control scheme seem to be simple validate of the existing predictive algorithm. The highlights of the work are unobvious.

Response:

We supplemented two evaluated metrics Accuracy and G-score, then we compared our model with the four competitors. Additionally, we conducted statistical analysis on these results. Please see Line #326 to Line #340.

4. The simulation process is too simple. Only bar charts are provided cannot be accepted.

Response:

We supplemented the experiments in Section 5.1. Our goal is to analyze the factors that affect athlete training performance of football athletes, therefore, we used the SPSS tool to analyze the relations between the seven training indicators and the forecast training performance. Please see Line #293 to Line #298.

To further observe the seven training indicators, we used k-means clustering to cluster the seven training indicators. Please see Line #308 to Line #314. Meanwhile, we visualized the clustered results in Fig. 3 in Line #319.

Additionally, we analyzed the training loss and testing loss of our model in Line #315 to Line #318. And please see the Fig. 4 in Line #324.

---

## [Editor Report · Decision Letter 2]

2 Jan 2025

Training forecast to football athletes using Hopfield neural networks based on Markov matrix

PONE-D-24-27336R2

Dear Dr. hongxing peng,

We’re pleased to inform you that your manuscript has been judged scientifically suitable for publication and will be formally accepted for publication once it meets all outstanding technical requirements.

Kind regards,

Perepi Rajarajeswari

Academic Editor

PLOS ONE

Additional Editor Comments (optional):

Now the paper is in accepted.
---

## [Editor Report · Acceptance letter]

PONE-D-24-27336R2

PLOS ONE

Dear Dr. peng,

I'm pleased to inform you that your manuscript has been deemed suitable for publication in PLOS ONE. Congratulations! Your manuscript is now being handed over to our production team.

Kind regards,

on behalf of

Dr. Perepi Rajarajeswari

Academic Editor

PLOS ONE